# Measurement of Serum IgG Anti-Integrin αvβ6 Autoantibodies Is a Promising Tool in the Diagnosis of Ulcerative Colitis

**DOI:** 10.3390/jcm11071881

**Published:** 2022-03-28

**Authors:** Niclas Rydell, Helena Ekoff, Per M. Hellström, Robert Movérare

**Affiliations:** 1Thermo Fisher Scientific, SE-75137 Uppsala, Sweden; niclas.rydell@thermofisher.com (N.R.); helena.ekoff@thermofisher.com (H.E.); 2Department of Medical Sciences, Gastroenterology and Hepatology, Uppsala University, SE-75185 Uppsala, Sweden; per.hellstrom@medsci.uu.se; 3Department of Medical Sciences, Respiratory, Allergy and Sleep Research, Uppsala University, SE-75185 Uppsala, Sweden

**Keywords:** autoantibody, diagnosis, inflammatory bowel disease, integrin αvβ6, ulcerative colitis

## Abstract

IgG anti-integrin αvβ6 autoantibodies (IgG anti-αvβ6) have been described as highly sensitive and specific markers of ulcerative colitis (UC) in the sera of Japanese inflammatory bowel disease (IBD) patients. We aimed to evaluate the diagnostic performance of IgG anti-αvβ6 as a biomarker in Swedish patients with IBD or irritable bowel syndrome (IBS). The study included adult UC (*n* = 59), Crohn’s disease (CD, *n* = 38), and IBS patients (*n* = 100). Partial Mayo score and Harvey–Bradshaw index were used to assess disease severity for UC and CD, respectively. Serum levels of IgG anti-αvβ6, reported as absorbance units (AU), were measured using an in-house ELISA where the 95th percentile of 76 healthy controls defined positivity. Faecal calprotectin (fCP) was measured using a commercial assay. The majority of the IBD patients were on medical treatment, and many were in remission (UC: 40.7%; CD: 47.4%). Seventy-one percent of the UC patients, 74.2% of CD patients, and 23.1% of the IBS patients had fCP test results >50 mg/kg. The UC group had significantly higher IgG anti-αvβ6 levels (median: 1.76 AU) than the CD and IBS groups (0.34 and 0.31 AU, both *p* < 0.0001). The diagnostic sensitivity of IgG anti-αvβ6 in UC was 76.3%, and the specificities were 79.0% (vs. CD) and 96.0% (vs. IBS). The IgG anti-αvβ6 levels related to disease severity of the UC patients (*p* < 0.01–0.05). Our study shows that IgG anti-αvβ6 is associated with UC in Swedish IBD patients and that the levels of the autoantibodies reflect disease severity. IgG anti-αvβ6 could be an attractive complement to fCP in the diagnostic work up of IBD patients.

## 1. Introduction

Ulcerative colitis (UC) is a chronic inflammatory condition affecting the colon and rectum. It is one of the two major forms of inflammatory bowel disease (IBD), the other one being Crohn’s disease (CD) [1]. The cause of UC is not known, and several theories have been proposed, which include environmental factors, dysregulation of the immune response, dysbiosis, and genetic factors [2]. Dysfunction of the intestinal epithelial barrier may lead to inflammatory cascades inducing a chronic state of the condition, or conversely, inflammatory mediators may disrupt the barrier, which may propagate the inflammation [1].

There is no single gold-standard method for the diagnosis of UC, which means the diagnosis is commonly made through multiple modalities including endoscopy, histopathology, imaging and laboratory testing, as well as absence of alternative diagnoses [3,4]. Laboratory testing includes elevated C-reactive protein (CRP), anaemia, hypoalbuminaemia, and faecal biomarkers such as faecal calprotectin (fCP), the latter being frequently used to distinguish irritable bowel syndrome (IBS) from IBD [5] as well as monitoring treatment response and disease activity [6,7,8].

Integrins are cell surface glycoprotein receptors composed of α- and β-subunits. The integrins are involved in cell signalling, proliferation, cell adhesion, and migration [9]. Integrins may play a central role in the pathogenesis of UC, and there is clinical evidence suggesting a positive benefit of integrin-blocking, preventing integrin-mediated intestinal homing of lymphocytes [10]. Integrin αvβ6 seems to be restricted to epithelial cells [11] and has been reported to play a part in maintaining epithelial barrier functions [12,13]. Furthermore, integrin αvβ6 can activate transforming growth factor-β1 to modulate innate immune surveillance along the gastrointestinal tract [14].

Kuwada et al. recently showed that a majority of UC patients in a Japanese cohort had IgG autoantibodies against integrin αvβ6 (IgG anti-integrin αvβ6) and that these autoantibodies may be highly sensitive and specific markers of UC [15].

The aim of the present study was to evaluate the diagnostic performance of serum IgG anti-integrin αvβ6 in Swedish IBD patients, diagnosed with UC or CD, and patients with IBS.

## 2. Materials and Methods

### 2.1. Patients and Study Design

Adult patients diagnosed with IBD (UC, *n* = 59; CD, *n* = 38) or IBS (*n* = 100) were consecutively asked to participate in the cross-sectional study performed at the outpatient Gastroenterology Clinic, Uppsala University Hospital between August 2017 and January 2021. Inclusion criteria of the study was IBD patients in clinical remission as well as with active disease. Exclusion criteria were cancer diagnosis, intestinal surgery within 6 months from the sample collection, or ongoing gastrointestinal infections. The IBD patients were previously clinically diagnosed according to a combination of symptoms, endoscopic finding, and histopathology as well as absence of alternative diagnoses [3,4]. Their median disease duration was 10 years (range: 0.5–57 years). The IBS patients used for comparison were diagnosed according to the Rome IV criteria [16]. All patients presenting with symptoms of IBS had a full blood count, C-reactive protein (CRP), celiac serology, and in patients below 45 years of age with diarrhoea, a faecal calprotectin. Venous serum samples were collected from all patients and stored at −70 °C until analysis. The patients were requested to collect a stool sample at home and deliver it to the clinic. The stool samples were stored at −70 °C until extraction and subsequent calprotectin analysis.

The currently used medication was retrieved from the electronic medical records of each patient. IBS patients were not treated with any anti-inflammatory, antibiotic, immunosuppressive, or biological medication. Disease activity at the time point for serum sampling was determined using the partial Mayo score for UC [17], the Harvey–Bradshaw index for CD [18], and for the IBS patients, the IBS symptom severity score was used [19].

Table 1 shows the characteristics of the patients. In addition, sera from 76 common blood donors, purchased from the German Red Cross, were used as healthy controls (HC) to calculate the cut-off for positive results of the IgG anti-integrin αvβ6 assay.

Clinical characteristics of the IBD and IBS patients are summarised in Appendix A.

### 2.2. IgG Anti-Integrin αvβ6

An enzyme-linked immunosorbent assay (ELISA) for the detection of serum IgG autoantibodies against integrin αvβ6 was developed based on the ELISA described by Kuwada et al. [15]. Briefly, microtiter plates were coated with 1 µg/mL of recombinant integrin αvβ6 (R&D Systems, Minneapolis, MN, USA) in 100 µL 0.1 M Na-carbonate buffer at pH 9.5 and allowed to incubate overnight at room temperature. The plates were washed with 0.9% saline containing 0.02% (*v*/*v*) Tween 20 and subsequently blocked with phosphate-buffered saline (PBS) containing 1% bovine serum albumin for 2 h at 37 °C. Then, they were again washed with 0.9% saline containing 0.02% (*v*/*v*) Tween 20 and incubated with 100 µL of patient sera diluted 1:150 in PBS with 0.5% (*v*/*v*) Tween 20 for 60 min at 37 °C. After washing, the plates were incubated with 100 µL rabbit polyclonal anti-human IgG antibody (Agilent, Santa Clara, CA, USA) conjugated with horseradish peroxidase (HRP) diluted 1:20,000 in PBS with 0.5% (*v*/*v*) Tween 20 at 37 °C for 60 min. After washing, the bound reactants were detected by incubation for 15 min with 3,3′, 5,5′- tetramethylbenzidine (BD, Franklin Lakes, NJ, USA), which was stopped with 100 µL 0.16 M sulphuric acid. All buffers except substrate and stop solution were supplemented with 1 mM each of MgCl_2_ and CaCl_2_ to increase integrin stability, as described by Kuwada et al. [15]. Absorbance units (AU) measurement was noted at 450 nm. Cut-off for positive results was defined as the 95th percentile of the AU values of 76 Red Cross blood donors (i.e., the HC subjects).

### 2.3. C-Reactive Protein

C-reactive protein (CRP) was analysed at the Laboratory of Clinical chemistry, Uppsala University Hospital on a Mindray BS380 (Mindray, Shenzhen, China) chemistry analyser using reagents (6K26-41) and calibrators (6K26-10) from Abbott Laboratories (Abbott Park, IL, USA). The testing was performed blinded without clinical information [20].

### 2.4. Faecal Calprotectin

Faecal calprotectin (fCP) was determined using EliA^™^ Calprotectin 2 (Thermo Fisher Scientific, Uppsala, Sweden), using a Phadia^™^ 250 instrument according to the manufacturer’s instructions. Cut-off for positive results was 50 mg/kg according to the manufacturer’s instructions. Values below the lower limit of the measuring range (3.8 mg/kg) were assigned an arbitrary value of 1.9 mg/kg.

### 2.5. Statistical Analysis

Sensitivity, specificity, positive predictive value (PPV), negative predictive value (NPV), likelihood ratio, and odds ratios were calculated. The Mann–Whitney U test and Fisher’s exact test (both two-tailed) were used for pair-wise comparisons of continuous parameters and categorical data between groups, respectively. Correlations between the biomarkers were calculated using the Spearman’s rank correlation coefficient (r). All *p* values < 0.05 were considered statistically significant. Receiver operating characteristic (ROC) analysis was used to evaluate the performance of IgG anti-integrin αvβ6 for diagnosis of UC vs. CD, IBS, and HC by calculating the area under curve (AUC). All statistical analyses were performed using GraphPad Prism version 8.1.2 for Windows (GraphPad Software, San Diego, CA, USA).

## 3. Results

### 3.1. Patient Characteristics

The UC, CD, and IBS patients had similar age but with significantly more females in the IBS group compared to the UC and CD groups (Table 1). The majority of the IBD patients were on medical treatment, and many were in remission. The UC and CD patients had significantly higher CRP than the IBS patients. Stool samples were obtained from 86.3% of the IBD and IBS patients. The single reason for lack of fCP data was absence of stool samples for analysis. Thirty-four (70.8%) of the UC patients, 23 (74.2%) of the CD patients, and 21 (23.1%) of the IBS patients had a positive fCP test result. The fCP levels were significantly higher for both the UC and the CD groups compared to the IBS group (Table 1). There was no significant difference in fCP levels between UC and CD. Five of the IBS patients had elevated fCP concentrations ranging from 213 to 523 mg/kg, and three had elevated CRP ranging from 10.7 to 25.8 mg/L (details in Appendix A). There were no correlations between fCP, CRP, or the IgG anti-integrin αvβ6 autoantibodies for these eight patients.

### 3.2. IgG Anti-Integrin αvβ6

IgG autoantibodies against integrin αvβ6 above the defined cut-off (AU 1.0, Figure 1) were predominantly observed in the UC group, in which 45 patients (76.3%) were positive in the ELISA. Autoantibody levels above the cut-off were also detected in serum from eight patients (21.1%) in the CD group. Even if the IgG anti-integrin αvβ6 levels apparently were high in some of these CD samples, the difference between the UC and CD groups was highly significant (Figure 1). Four patients (4.0%) in the IBS group had levels of autoantibodies above the cut-off. None of these patients had elevated CRP or fCP (stool samples were missing for two of these patients). No differences due to variations in gender distributions were observed for IgG anti-integrin αvβ6 levels in any of the groups (data not shown).

The best diagnostic performance of IgG anti-integrin αvβ6 evaluated by ROC curve analysis was almost equal for UC vs. IBS with an AUC of 0.945, and UC vs. HC with an AUC of 0.946 (Appendix A). Thus, a clinical sensitivity of 76.3% and a specificity of 96.0% was obtained for diagnosis of UC vs. IBS, and for differentiation from CD the specificity of the IgG anti-integrin αvβ6 ELISA reached 79.0% (Table 2).

There was an association between the level of IgG anti-integrin αvβ6 autoantibodies and disease severity in the UC group. The patients having mild, moderate, or severe disease activity according to the partial Mayo score had significantly higher levels of IgG anti-integrin αvβ6 than UC patients in remission (Figure 2). UC patients in remission had significantly higher IgG anti-integrin αvβ6 levels than CD patients in remission (*p* = 0.0027), and UC patients with active disease (pooled patients with mild, moderate, and severe disease) also had significantly higher IgG anti-integrin αvβ6 levels than CD patients with active disease (*p* < 0.0001) (data not shown).

There was no significant association between the level of IgG anti-integrin αvβ6 and the disease activity for the CD patients, and there was no correlation between the level of IgG anti-integrin αvβ6 and disease years neither for the UC nor the CD patients (data not shown).

A correlation (Spearman r = 0.467, *p* < 0.001) was found between the levels of IgG anti-integrin αvβ6 and CRP for the UC patients, but not for the CD patients or the whole IBD group (UC + CD) (Table 3). Correlations were also observed between IgG anti-integrin αvβ6 and fCP for the UC patients and the whole IBD group (UC + CD) (Table 3). CRP correlated with fCP for the UC patients (r = 0.495, *p* < 0.001) and the whole IBD group (r = 0.364, *p* < 0.01), but not for the CD patients.

## 4. Discussion

We have shown that the measurement of serum IgG autoantibodies against integrin αvβ6 could be a valuable tool for the diagnosis of UC. The level of autoantibodies also reflects the disease severity in UC patients. To the best of our knowledge, this is the first study in a European cohort that can confirm the results from the Japanese study in which the autoantibodies originally were described [15]. In the study by Kuwada et al., it was shown that colonic epithelial integrin αvβ6 was a putative autoantigen in UC and that the autoantibodies in UC patients could block the binding of integrin αvβ6 to fibronectin in connective tissue.

By using an ELISA coated with integrin αvβ6, Kuwada et al. showed an increased antibody binding when they added Ca^2+^ and Mg^2+^ to the assay buffers [15]. We observed a similar dependency of the ions in our ELISA (data not shown) and the buffers were supplemented with 1 mM MgCl_2_ and 1 mM CaCl_2_.

The study by Kuwada et al. included Japanese IBD patients and healthy controls but did not include IBS patients. They reached a sensitivity of 87.5% and a specificity of 95.0% in their validation group, which in comparison to the present study was a bit higher. Our study including Swedish adult patients showed a sensitivity of 76.3% for IgG anti-integrin αvβ6 in diagnosis of UC together with specificities of 79.0% (vs. CD) and 96.0% (vs. IBS). The differences between our study and the Kuwada study might be due to dissimilarities in assay optimization and definition of diagnostic cut-off, and/or genetic or biological variations between the cohorts. In addition, the disease diagnostic criteria for UC may have slightly differed between the sites.

The high sensitivity and specificity of the IgG anti-integrin αvβ6 measurement could be considered a major step forward toward an objective parameter in the aid of the diagnosis of UC and possibly also for monitoring of disease activity and treatment response using an undemanding laboratory test that could be easily automated.

A limitation of the present study is that a rather large portion of the IBD patients were in remission (UC: 40.7%, CD: 47.4%), and most of the patients had lived a long time with their disease (median: 10 years, range: 0.5–57 years) as opposed to being newly diagnosed or treatment naïve. Even if there was a clear-cut correlation between the IgG anti-integrin αvβ6 autoantibody and disease activity of UC, we found no correlation between the levels of IgG anti-integrin αvβ6 and disease years. It would be interesting to see the autoantibody levels in newly diagnosed and treatment naïve IBD-patients and follow them longitudinally. The low number of patients in the CD group also made statistical evaluation of the effect of disease severity regarding the IgG anti-integrin αvβ6 difficult for this group, and some effects might have been missed because of that. Furthermore, endoscopic information on disease location at the time of the blood sampling was not available, and therefore, it cannot be ruled out that results of the study could be explained by disease location rather than disease type. It would be highly interesting to investigate the levels of IgG anti-integrin αvβ6 in relation to disease location, and further investigations are warranted in this area. Another limitation of the present study is the lack of a validation cohort, but nonetheless, the results are encouraging and highly corroborative with the ones from Kuwada et al. [15].

Faecal calprotectin is a standard test used to distinguish IBD from IBS with a reported diagnostic sensitivity and specificity of 75–83% and 68–95%, respectively [21]. As previously mentioned, a majority of the IBD patients were undergoing medical treatment, and many of them were in remission, which hampers an accurate evaluation of the diagnostic performance of fCP in comparison with IgG anti-integrin αvβ6. However, there are known patient compliance issues with stool samples due to an obvious lack of esthetical appeal. It is considered both awkward and complicated for the patients to handle faecal samples, which require a certain degree of motivation. There are logistical aspects of sending in the samples or bringing them to the doctor’s office in person, as well as forgetfulness, accounting for compliance rarely exceeding 60% of the patients [22,23,24]. Although the collection of blood is an invasive procedure, it is usually accepted by most patients, and it is swiftly and easily performed in conjunction with routine or follow-up visits. In this study, 13.7% of the IBD and IBS patients did not comply regarding the provision of stool samples despite being reminded at least once, whereas 100% of the patients complied with venous serum sampling.

Serum as sample matrix is very homogenous as opposed to stool samples that may vary vastly regarding their consistency and usually require cumbersome preanalytical preparation steps. As a result of this, fCP is burdened with some degree of uncertainty especially regarding the intra-individual variability of samples taken only a few days apart [25]. This is generally not a problem for antibody measurements in serum samples.

From a compliance perspective, it corroborates the importance of finding biomarkers that can be analysed in the blood rather than in stool, not needing to consider the inherent variability of stool samples.

The five IBS patients with the highest fCP concentrations had levels ranging from 213 to 523 mg/kg. There is no obvious explanation for this, as four of them had normal endoscopies (one was not endoscopically examined). Neither of these patients were obese (all had body mass index <30) nor did they show elevated CRP (all <5 mg/L). In the IBS group, 21/91 (23%) had fCP >50 mg/kg and 11/91 (12%) had fCP >100 mg/kg. However, a recent study showed that 34% of patients with normal findings on their colonoscopy had elevated levels of fCP (>50 mg/kg), and high fCP was not associated with an increased risk for future development of IBD or other gastrointestinal diseases during the 3-year follow-up period of the study [26].

Although the association of IgG anti-integrin αvβ6 with disease severity in UC shown in the present study warrants further research, the results are promising. A similar association was also presented by Kuwada et al. [15]. The disease activity is commonly monitored by patient symptoms and laboratory tests such as CRP, erythrocyte sedimentation rate, or other non-specific biological markers and may therefore result in the need for multiple colonoscopies [3,13,27]. The measurement of IgG anti-integrin αvβ6 autoantibodies could be an acceptable and well-tolerated tool for the diagnosing and monitoring of UC disease activity.

In conclusion, we have shown that the measurement of IgG anti-integrin αvβ6 could be a valuable tool in the diagnosis of UC and that the levels of the autoantibodies also may reflect disease severity. Considering the common lack of compliance regarding stool sampling and fCP for screening of IBD, an IgG anti-integrin αvβ6 test could be an attractive complement to fCP in the diagnostic work up of IBD and IBS patients.

## Figures and Tables

**Figure 1 jcm-11-01881-f001:**
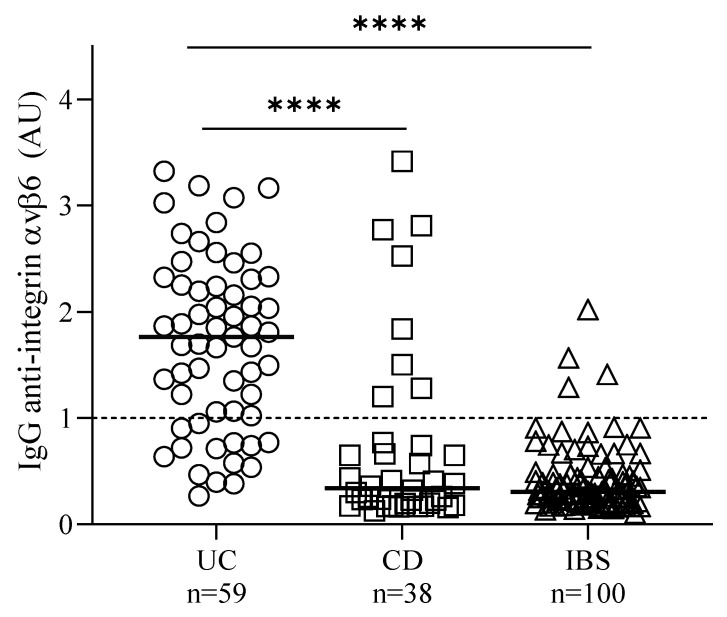
Serum IgG anti-integrin αvβ6 autoantibodies in patients with ulcerative colitis (UC), Crohn’s disease (CD), and irritable bowel syndrome (IBS). The dashed line indicates the assay cut-off level based on the 95th percentile for the HC group. Median and significant differences between the groups are indicated (**** *p* < 0.0001).

**Figure 2 jcm-11-01881-f002:**
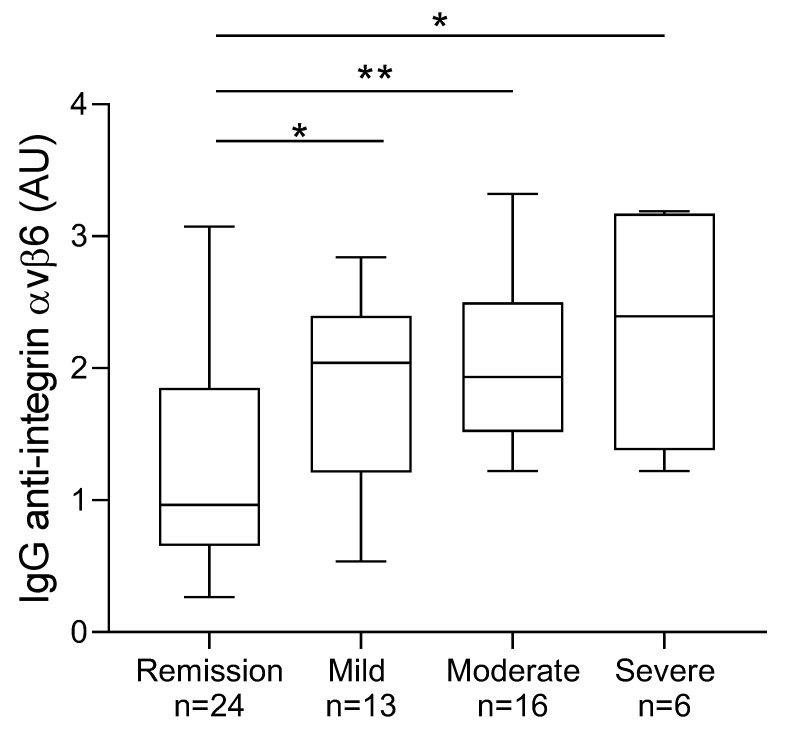
IgG anti-integrin αvβ6 autoantibody concentration vs. ulcerative colitis disease severity according to the partial Mayo score. Significant differences between the severity groups are indicated (* *p* < 0.05, ** *p* < 0.01).

**Table 1 jcm-11-01881-t001:** Characteristics of patients.

	UC Group	CD Group	IBS Group
Number of subjects	59	38	100
Age (years, median with min–max range)	38 (18–79)	39 (20–78)	36 (18–57)
Gender (% male)	55.9	47.4	29.0 ^(1)^
CRP (mg/L, median with min–max range)	2.25 (1.47–18.2)	2.52 (1.40–137)	2.05 (1.43–25.8) ^(2)^
fCP (mg/kg, median with min–max range) ^(3)^	306 (1.90–4207) ^(4)^	166 (9.60–14,028) ^(4)^	16.0 (1.90–524)
Disease activity ^(5)^			
Remission, *n* (%)	24 (40.7)	18 (47.4)	n/a
Mild disease, *n* (%)	13 (22.0)	11 (28.9)	10 (10)
Moderate disease, *n* (%)	16 (27.1)	8 (21.1)	37 (37)
Severe disease, *n* (%)	6 (10.2)	1 (2.63)	52 (52)
Medicated IBD patients (%)	83.0	73.7	n/a
Treatment			
Steroids, *n* (%)	13 (22.0)	9 (23.7)	n/a
Immunosuppressives, *n* (%)	12 (20.3)	16 (42.1)	n/a
5-ASA, *n* (%)	38 (64.4)	8 (21.1)	n/a
Antimetabolites, *n* (%)	1 (1.69)	0 (0)	n/a
Biologics, *n* (%)	7 (11.9)	12 (31.6)	n/a
No medication, *n* (%)	9 (15.3)	10 (26.3)	n/a

UC, ulcerative colitis; CD, Crohn’s disease; IBS, irritable bowel syndrome; n/a, not applicable. ^(1)^ Significantly fewer males than the UC group (*p* < 0.0001). ^(2)^ Significantly lower than the UC group (*p* < 0.05) and CD group (*p* < 0.01). ^(3)^ Number of faecal samples provided: UC (*n* = 48), CD (*n* = 31), IBS (*n* = 91). ^(4)^ Significantly higher than the IBS group (*p* < 0.0001). ^(5)^ UC: partial Mayo score; CD: Harvey–Bradshaw index; IBS: IBS symptom severity score.

**Table 2 jcm-11-01881-t002:** Diagnostic performance of serum IgG anti-integrin αvβ6 autoantibodies in patients with UC and CD.

	Sens.	Spec.	PPV	NPV	OR (CI)	LR	*p* Value
UC vs. CD	0.763	0.790	0.849	0.682	12.1 (4.42–33.8)	3.62	<0.0001
UC vs. IBS	0.763	0.960	0.918	0.873	77.1 (23.2–214)	19.1	<0.0001
UC vs. HC	0.763	0.961	0.938	0.839	78.2 (22.0–253)	19.3	<0.0001
CD vs. IBS	0.211	0.960	0.667	0.762	6.40 (2.00–19.9)	5.26	0.0036
CD vs. HC	0.211	0.961	0.727	0.709	6.49 (1.75–23.4)	5.33	0.0053
IBD vs. IBS	0.546	0.960	0.930	0.686	28.9 (9.92–77.3)	13.7	<0.0001
IBD vs. HC	0.546	0.961	0.946	0.624	29.3 (9.13–92.5)	13.8	<0.0001

UC, ulcerative colitis; CD, Crohn’s disease; IBS, irritable bowel syndrome; HC, healthy controls; IBD, inflammatory bowel disease; Sens., sensitivity; Spec., specificity; PPV, positive predictive value; NPV, negative predictive value; OR, odds ratio; CI, confidence interval; LR, likelihood ratio.

**Table 3 jcm-11-01881-t003:** Correlation between IgG anti-integrin αvβ6 autoantibodies and C-reactive protein and faecal calprotectin.

	IgG Anti-Integrin αvβ6 vs. CRP		IgG Anti-Integrin αvβ6 vs. fCP	
	Spearman r	*p* Value	Spearman r	*p* Value
IBD	0.178	0.0820	0.309	0.0056
UC	0.467	0.0002	0.342	0.0175
CD	−0.0136	0.9356	0.196	0.2912

CRP, C-reactive protein; fCP, faecal calprotectin; IBD, inflammatory bowel disease; UC, ulcerative colitis; CD, Crohn’s disease.

## Data Availability

Individual patient information is included as a data supplement available with the online version of this article. Other data that support the findings of this study are available on request from the corresponding author for a period of 5 years after the publication date.

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
