# Peer review of "Measurement of Serum IgG Anti-Integrin αvβ6 Autoantibodies Is a Promising Tool in the Diagnosis of Ulcerative Colitis"

_jcm, 2022, doi:10.3390/jcm11071881_

Round 1

Reviewer 1 Report

Authors conducted a study in Swedish patients to validate previous results showing the interest of serum autoantibodies (IgG anti–integrin αvβ6) as a diagnostic marker of ulcerative colitis (UC) in Japanese cohort. The search of non-invasive tools to help the diagnosis of UC is a relevant question. Here, authors compared the serum level of IgG anti–integrin αvβ6 in UC, Crohn’s disease (CD) and irritable bowel syndrome (IBS) patients. The results seemed to confirm the interest of IgG anti–integrin αvβ6 for the diagnosis and monitoring of UC. Overall the study is well conducted, the results are interesting and the discussion is very relevant. I would have 2 major remarks/questions and some minor points:

Majors:

-Authors clearly showed that the serum level of IgG anti–integrin αvβ6 is different between UC and CD patients. However, there is less patients in remission for CD (51.3%) than for UC (38.7%) and it seems that faecal calprotectin is higher in UC than CD patients. How to prove that the higher serum level of IgG anti–integrin αvβ6 in UC than CD patients is not only due differences in disease activity rather than IBD types? In addition to the results presented, I suggest to separate the comparison (UC vs CD) for patients in remission and patients with active disease (pool mild, moderate and severe disease). If these 2 comparisons still show differences between UC and CD patients it will strengthen the results of the study (results independent from disease activity).

-Authors clearly showed that the serum level of IgG anti–integrin αvβ6 is different between UC and CD patients. However, they did not show the results of CD patients according to disease location. Indeed, ileal and colonic CD shows major differences (see reviews: doi:10.1016/j.cgh.2019.04.040, doi: 10.1038/s41575-021-00424-6, doi: 10.1111/apt.16536). Regarding biomarkers, data showed that the level of faecal calprotectin is higher in active colonic than active ileal CD. If this is also the case for the serum level of IgG anti–integrin αvβ6 it could change the message of the study. In this scenario, IgG anti–integrin αvβ6 could be more a marker of disease location rather than IBD types. Do authors can comment this point and provides additional data? Do authors have information on disease location, i.e., Montreal classification or even better endoscopic information on disease location at time of blood sampling? For instance, it would be interesting to know the location and disease activity of the 8 CD patients positive for IgG anti–integrin αvβ6.

If not presented (but justified), the relation between CD location and serum level of autoantibodies should be at least acknowledged as a limitation of the study. To my opinion, this will constitute a progress compared to the study of Kuwada et al which did not consider CD location in the analysis and discussion.

Minors:

-The chosen cut-off for the faecal calprotectin should be given in the abstract.

-It would be better to group the initial and final count of patients (lines 64-66 and 82-84)

-The medication of patients should be detailed in the Table 1 with categories: biologics, anti-metabolites, antibiotics, 5-ASA…

-“Median with range” in the Table 1 means interquartile range (IQR) or min-max (it seems min-max)? This should be clarified.

-Lines 154-155: “IgG autoantibodies against integrin avß6 were predominantly observed in the UC”. It let think that autoantibodies are not detected in CD patients. The sentence is not really correct.

-Lines 155-156: “The autoantibodies were also detected in serum from 8 patients (21.6%) in the CD group”. It let think that autoantibodies are not detected in all CD patients. The sentence is not really correct.

-Lines 158-159: “Four patients (4.0%) in the IBS group had low levels of autoantibodies. It should be “high” instead of “low” ?

-The cut-off presented in the Figure 1 should be also defined in the legend.

Again, congratulation for your study

Author Response

Dear Reviewer 1, Please see the attachment.

Reviewer 2 Report

The study by Rydell et al examines αVβ6 autoantibodies in Swedish patient with CD, UC and IBS. It is interesting and a well-executed first attempt to look at these in Caucasian patients. The findings are to be considered preliminary and require prospective study in patients undergoing 1st diagnostic work-up to determine their usefulness in the diagnostic process. The sue of IBS controls is very useful rather than heathy controls.

Please address the following points:

  1. Please confirm the work-up for the IBS group in the methods. We need to have more confidence these are truly IBS patients.
  2. Gender is not equally distributed amongst the groups could this be an influence on results?

3.The numbers for comparing disease activity within CD are too small and you may miss a potential effect here.

  1. The results should lead to a study where αVβ6 autoantibodies are examined in patients undergoing 1st diagnostic work-up of GI symptoms. This way we get a better idea how useful they are in this stage. The current study examined the levels in patients with 10 years disease duration only.
  2. Page 6 line 201. This sentence is too strongly worded given the lack of data regarding first diagnosis.

Author Response

Dear Reviewer 2, Please see the attachment.

Round 2

Reviewer 1 Report

Thank you your for your response. I have still one comment.

The limitation of the study regarding the effect of disease location is not well recognised (lines 244-249). The results of the study could be explained by disease location rather than disease types. To my opinion, this possibility needs to be recognised in order to provide a fair information for the readers.
